# *CodeChemist*: Test-Time Scaling for Low-Resource Code Generation via Functional Knowledge Transfer

**Kaixin Wang** [1] **Tianlin Li** [2] **Xiaoyu Zhang** [3] **Aishan Liu** [2] **Xianglong Liu** [2] **Ziqi Liu** [4] **Zhiqiang Zhang** [4]
**Jun Zhou** [4] **Bin Shi** [1 5]

## Abstract

Code Large Language Models (CodeLLMs) have been widely adopted for Natural Language to Programming Language code generation, powering applications with large user bases. Their performance, however, varies sharply across programming languages (PLs) and is particularly suboptimal for low-resource PLs due to data scarcity, limiting their overall usability. In this work, we introduce *CodeChemist*, a simple yet effective, training-free test-time scaling framework that transfers the model's functional knowledge from high-resource to low-resource PLs via synthesized test cases, without relying on external models. Specifically, *CodeChemist* first applies multi-temperature hedged sampling to generate a pool of candidate solutions in the low-resource PL and synthesizes a set of test inputs. It then estimates uncertainty: when uncertainty is low, it selects the output via in-language majority voting; otherwise, it constructs cross-lingual I/O test oracles by executing high-resource reference programs and selects the candidate with the highest pass rate. Extensive experiments demonstrate that *CodeChemist* significantly outperforms existing test-time scaling methods, improving code generation for both low-resource PLs (e.g., Lua) and complex-syntax PLs (e.g., C++, Java) without retraining.

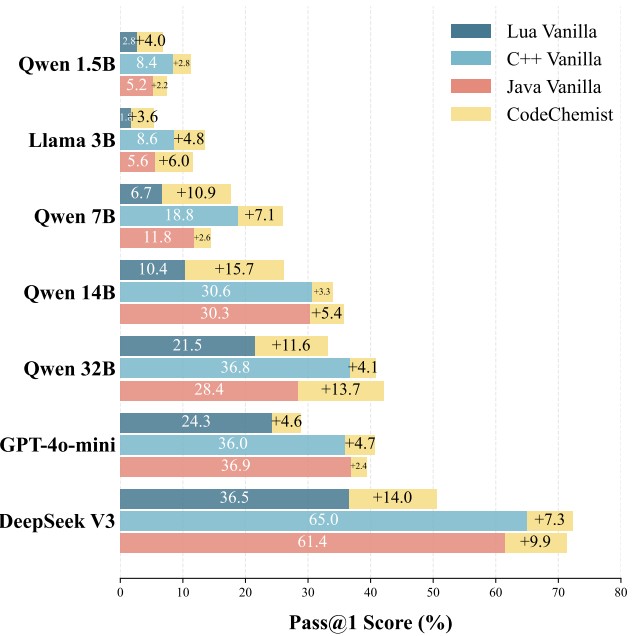

*Figure 1.* **Performance improvement with *CodeChemist* on Ag-LiveCodeBench-X (Boruch-Gruszecki et al., 2025).** *Code-Chemist* consistently enhances performance across a wide range of model sizes. The improvement is particularly pronounced for the low-resource language Lua. **Notably, DeepSeek-V3 enhanced by *CodeChemist* achieves 50.5 in Lua, surpassing the official leaderboard's top entry of 43.0 at the time of our evaluation**[1]. "Qwen" denotes the "Qwen2.5-Coder-Instruct" series.

## 1. Introduction

Large Language Models (LLMs) have catalyzed a transformative shift in code generation, driven by the emergence of specialized variants designed for programming tasks, referred to as Code Large Language Models (CodeLLMs). With powerful capabilities in code generation, these models have consistently outperformed traditional methods and are now extensively adopted in both academic and industrial settings (Hou et al., 2024; Wang et al., 2025a; Hui et al., 2024; Wang et al., 2025b). In particular, the capability of synthesizing code from Natural Language (NL) lays a critical foundation for intelligent programming. For example, widely used tools such as GitHub Copilot (GitHub, 2023), which leverage models like GPT-4 and Codex (Chen et al., 2021), allow users to generate code directly from NL instructions, greatly enhancing development efficiency.

---

[1]School of Computer Science and Technology, Xi'an Jiaotong University, Xi'an, China [2]Beihang University, Beijing, China [3]Nanyang Technological University, Singapore [4]Ant Group, Hangzhou, China [5]National Engineering Research Center for Visual Information and Applications, Xi'an, China. Correspondence to: Tianlin Li <tianlin001@buaa.edu.cn>, Bin Shi <shibin@xjtu.edu.cn>.

*Proceedings of the 43$^{rd}$ International Conference on Machine Learning*, Seoul, South Korea. PMLR 306, 2026. Copyright 2026 by the author(s).

---

[1]Leaderboard accessed on January 28, 2026. https://ag-livecodebench-x.github.io.

However, the performance of CodeLLMs in code generation varies significantly across programming languages (PLs). They excel in high-resource PLs like Python but underperform in low-resource PLs (e.g., Lua) or those with complex syntax (e.g., C++ and Java) (Zhang et al., 2024; Giagnorio et al., 2025; Cassano et al., 2024; Tarassow, 2023). This disparity limits the practical usability of CodeLLMs in multilingual development environments and hinders support for developers using less-represented PLs (Zheng et al., 2023b). Bridging this performance gap is vital to fully realize the potential of LLMs in real-world code generation applications.

The most straightforward way to improve performance in low-resource PLs is to collect additional training data and fine-tune the model. Considering the inherent data scarcity, several lines of research have turned to cross-lingual transfer techniques that leverage corpora from high-resource PLs. For instance, Roziere et al. (2022); Cassano et al. (2024) propose translating code snippets from high-resource into low-resource PLs. In practice, translated code snippets often suffer from limited quality, and the required training process is computationally expensive. As a result, the practicality of such methods is substantially constrained.

Recently, test-time scaling methods (Li et al., 2025a) have emerged as a promising alternative to costly training-based techniques for improving code generation. However, their gains on low-resource PLs are often limited because they overlook the fundamental challenge of data scarcity. Moreover, common remedies such as data augmentation are incompatible with the test-time paradigm, making low-resource improvements particularly challenging. Consequently, we pursue a test-time scaling strategy that transfers the model's inherent knowledge from high-resource PLs to improve performance in low-resource PLs.

In our paper, we propose *CodeChemist*[2], based on the key insight that test cases naturally encapsulate functional knowledge, which is the input-output-defined, PL-agnostic essence of a function's logic. Specifically, we first generate test cases from a high-resource PL, then produce diverse candidates in a low-resource PL using multi-temperature hedging, and select the one whose execution best matches the test cases. However, generating test cases from a high-resource language for every task to guide low-resource code generation is not only computationally expensive but can also be misleading. To mitigate this, we introduce an uncertainty-aware adaptive selection mechanism. Under this design, *CodeChemist* directly outputs the low-resource result when uncertainty is low.

We first conduct comprehensive experiments on Lua, a representative low-resource PL, across multiple models. As

---

[2]Code is available at https://github.com/whappily/CodeChemist.

shown in Figure 1, *CodeChemist* achieves remarkable performance gains. To further validate the extensibility, we evaluate it on PLs with complex syntax, namely C++ and Java. The results show that *CodeChemist* consistently improves performance across different PLs and models.

## 2. Related Work

### 2.1. Enhancing CodeLLMs for Low-resource PLs

CodeLLMs exhibit a significant performance gap between high-resource PLs (e.g., Python) and low-resource PLs, which has attracted considerable research attention. Existing approaches primarily focus on fine-tuning methods.

These fine-tuning methods are typically designed to curate additional data for low-resource PLs, which is then used to fine-tune a model and enhance its performance on them. (Chen et al., 2022b) propose selecting high-resource PLs for auxiliary training based on their similarity to a target low-resource PLs. A key limitation of this approach, however, is its high task-sensitivity and limited generalization. Another line of work follows a "translation–testing–filtering" paradigm. For instance, TransCoder-ST (Roziere et al., 2022) first translates code from a high-resource PL into a low-resource PL. It then constructs a fine-tuning dataset by filtering the translated samples for validity using automatically generated unit tests. However, generating these unit tests depends on language-specific toolchains. Since many low-resource PLs lack such toolchains, this approach is difficult to generalize. MultiPL-T (Cassano et al., 2024) improves upon this by generating unit tests through CodeLLMs only in high-resource PLs. It then translates both the code and its corresponding tests into the target low-resource PLs, using execution-based verification to build a reliable training dataset. However, its effectiveness is highly dependent on the quality of the LLM-based translation for both the code and the test cases. Furthermore, Bridge-C (Zhang et al., 2024) introduces a two-stage framework that first synthesizes an intermediate "code-bridge" (HRPL code with annotations) to guide the generation of LRPL data for alignment training. Even with high-quality synthetic datasets, these fine-tuning-based methods can impair the model's performance on high-resource PLs. Furthermore, mastering complex linguistic constructs remains challenging even with additional targeted low-resource data.

Different from the prior work, in this paper, we propose *CodeChemist* that transfers knowledge across PLs at inference time. This approach requires no extra training data and achieves higher performance through test case validation.

### 2.2. Test-Time Scaling

Test-time scaling is a technique used to enhance the reasoning capabilities of LLMs during inference by allocating

more computational resources. A widely used approach is to generate multiple candidate solutions and apply a selection mechanism to choose the most promising one, commonly known as Best-of-N sampling. Within this framework, common selection strategies include (weighted) majority voting (Wang et al., 2023a), automated judgment by an LLM (LLM Judge) (Wang et al., 2025c), and scoring with a trained reward model (Christiano et al., 2017; Lightman et al., 2023). However, these strategies often struggle to identify the truly best candidate (Stroebl et al., 2026; Brown et al., 2024; Hassid et al., 2024).

Test-time scaling has also shown great potential in enhancing code generation. CodeMonkeys (Ehrlich et al., 2025) is an approach that enhances the performance of LLMs in the SWE-bench benchmark by extending test-time compute. The system generates test scripts and uses execution feedback to continuously optimize candidate code snippets. After several iterations, it combines majority voting and model selection to choose the best solution. S* (Li et al., 2025a) is a hybrid test-time extension method that uses an external model to generate test inputs and then feeds execution feedback to the LLM for optimal selection. However, the application of these methods to low-resource PLs has been largely underexplored.

### 2.3. Enhancing Code Generation through Test Cases

Using synthetic test cases to guide code generation has emerged as an effective approach (Chen et al., 2022a; Huang et al., 2024; Jiao et al., 2025). (Lee et al., 2025) proposes an adversarial reinforcement learning framework that optimizes the test case generator and code generator through adversarial training, selecting the optimal code based on the number of test cases it passes. Similarly, (Zeng et al., 2025) trains a reward model by constructing a problem-test case dataset and then scores the candidate code snippet to select the optimal solution. GenX (Wang et al., 2024) jointly trains the code generation model and the test generation model through execution feedback, allowing them to improve each other over time. Epicoder (Wang et al., 2025d) introduces a data synthesis framework based on feature trees and relies on LLM-generated test files to iteratively debug code.

However, the above methods rely on the model to directly generate input-output pairs, but due to hallucinations, the model may introduce inaccuracies in predicting the correct outputs.

## 3. Methodology

We propose *CodeChemist*, an uncertainty-aware framework illustrated in Figure 2. Specifically, *CodeChemist* first applies multi-temperature hedged sampling to generate a diverse pool of candidate programs in the target PL, and gen-

erates a set of test inputs. It then estimates the model's uncertainty. If the generation has low uncertainty, *CodeChemist* selects the low-resource output via majority voting. Otherwise, under high uncertainty, *CodeChemist* performs cross-lingual knowledge transfer.

### 3.1. Problem Definition

We focus on the task of Natural Language to PL (NL→PL) generation. Formally, let $\mathcal{P}$ denote a programming problem description in natural language. Our objective is to employ a model $\mathcal{M}$ to generate a code solution $y$ in a target PL $\mathcal{L}$.

### 3.2. Hedged Sampling

The sampling stage aims to produce a pool of candidate code snippets in the target PLs that balances quality with diversity, thereby ensuring a sufficiently rich solution space for the subsequent selection stage. The key challenge lies in temperature configuration, as it directly controls the diversity-quality trade-off and must be carefully calibrated.

In standard sampling, the temperature parameter $\tau$ controls the smoothness of the softmax distribution, thereby influencing the diversity and determinism of the generated samples. For a given temperature $\tau_j$, the probability of selecting token $v_k$ is:

$$P_{\tau_j}(v_k) = \frac{\exp(l_k/\tau_j)}{\sum_i \exp(l_i/\tau_j)}. \tag{1}$$

$\tau$ regulates the trade-off between diversity and quality (Ye et al., 2025). When $\tau$ is large, the generated samples become more diverse. As $\tau \to 0$, the distribution sharpens and the results become deterministic. At $\tau = 0$, it corresponds to greedy decoding.

Configuring the temperature parameter $\tau$ for low-resource PLs is challenging due to two primary factors. ❶ **Limited Confidence in Low-Resource PLs.** Due to limited and often lower-quality training data, low-resource PLs tend to produce "flat" output distributions, in contrast to the confident predictions typical of high-resource PLs. ❷ **Context-Dependent Optimality.** The optimal $\tau$ is highly context-dependent, varying significantly across models, tasks, and languages since each occupies distinct subspaces of the training distribution (Li et al., 2025b). This results in a combinatorial explosion over the combinations of model, dataset, and language, making fine-grained $\tau$ tuning prohibitively expensive and impractical for real-world applications.

Based on the above considerations, and motivated by the language-agnostic benefits of diversified sampling (Khairi et al., 2025), we adopt a multi-temperature hedged sampling strategy to generate a candidate pool of low-resource program code. This method is designed to be universally ap-

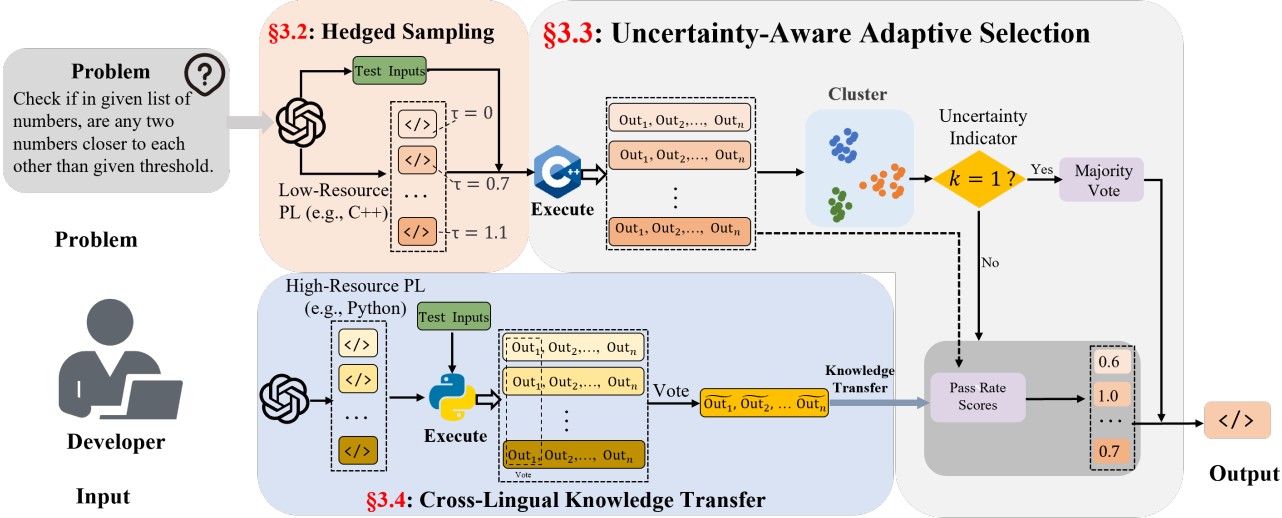

*Figure 2.* The overview of *CodeChemist*. $k$ represents the uncertainty of the target PL candidate pool.

plicable across PLs, balancing quality and diversity. Specifically, we draw samples using multiple high-temperature values (to encourage diversity) while also including the greedy-decoding sample ($\tau = 0$). For instance, we selected temperatures of 0, 0.7, 0.9, and 1.1, with the number of samples being 1, 3, 3, and 3, respectively.

### 3.3. Uncertainty-Aware Adaptive Selection

Given a candidate pool $Y$ generated via hedged sampling and the test inputs $I$ (prompted from CodeLLMs), we estimate the model's uncertainty and adaptively decide whether to invoke cross-lingual knowledge transfer.

We consider uncertainty estimation from two complementary perspectives: semantic and syntactic. This is because, while execution-based clustering (Chen et al., 2022a; Li et al., 2022) captures functional consensus, it can be misled by "spurious consistency"—cases where incorrect implementations coincidentally produce identical outputs on $I$. We therefore incorporate Abstract Syntax Tree (AST) based clustering to reduce the risk of such spurious consensus. Specifically, we execute the candidates on the input set $I$ and cluster the pool based on execution results and AST similarity (Zhang & Shasha, 1989). We estimate uncertainty based on the number of clusters. Let $k_{\text{ext}}$ and $k_{\text{ast}}$ denote the number of distinct execution clusters and AST clusters, respectively. We introduce tolerance thresholds $\delta_{ext}$ and $\delta_{ast}$, and define the uncertainty indicator $\mathcal{K}$:

$$\mathcal{K}(\delta_{ext}, \delta_{ast}) = \mathbb{I}(k_{ext} \leq \delta_{ext}) \cdot \mathbb{I}(k_{ast} \leq \delta_{ast}). \quad (2)$$

When $\mathcal{K} = 1$, the model exhibits high internal consistency. In this regime, we select the final output by majority voting within the dominant cluster, avoiding any additional oracle

construction cost. When $\mathcal{K} = 0$, indicating high uncertainty, we proceed to cross-lingual knowledge transfer by constructing I/O test oracles using the same test inputs $I$, as described in subsection 3.4.

### 3.4. Cross-Lingual Knowledge Transfer

When cross-lingual knowledge transfer is triggered, we construct test cases as input/output (I/O) test oracles based on a high-resource PL. We first generate $h$ reference implementations $H$ in a high-resource PL and execute them on each input $i_j \in I$. Executions resulting in compilation failures, timeouts, or crashes are discarded. From the remaining successful runs, we determine the oracle output for each input by majority voting, yielding an I/O test oracle set $T = \{t_1, \ldots, t_M\}$. We apply a unified Serialization Protocol (see Appendix A.2) to normalize these outputs for cross-lingual comparison (e.g., unifying C++'s `true` and Python's `True`).

Given the oracle set $T$, we evaluate each low-resource candidate $y \in Y$ by its pass rate:

$$S(y) = \frac{1}{M} \sum_{i=1}^{M} \text{pass}(y, t_i), \quad (3)$$

where $\text{pass}(y, t_i) = 1$ if candidate $y$ matches the oracle output on test oracle $t_i$; otherwise (including compilation failure, timeout, or output mismatch) $\text{pass}(y, t_i) = 0$. We select the final output as:

$$\hat{y} = \arg, \max_{y \in Y} S(y). \quad (4)$$

To ensure robustness, if all candidates fail (i.e., $\max S(y) = 0$), we fall back to the greedy sample. In the event of a

tie, candidates generated with lower sampling temperatures are prioritized. The pseudocode for the implementation of *CodeChemist* is presented in Appendix A.1.

## 4. Experiments

In this section, we conduct comprehensive experiments to evaluate the effectiveness of *CodeChemist* across multiple model families, different model sizes, and various benchmark difficulties.

### 4.1. Experiment Setup

The experimental setup includes metrics, model selection, benchmarks, baselines, and implementation details.

**Metrics.** We evaluate performance using the Pass@1 estimator proposed by Chen et al. (2021) with $n = 10$ samples per problem.

**Models.** To comprehensively evaluate the performance of *CodeChemist* across models of different sizes, we select multiple variants from the same model series. Specifically, we choose the Qwen2.5-Coder-Instruct (Hui et al., 2024) (referred to as Qwen) series (including the 1.5B, 3B, 7B, 14B, and 32B versions), Llama3.2 (Dubey et al., 2024) (3B version), GPT-4o mini (Hurst et al., 2024) (referred to as 4o-mini), and introduce the DeepSeek-V3-chat (Liu et al., 2024) (referred to as DeepSeek) model for comparison.

**Benchmarks.** We use MultiPL-E (Cassano et al., 2022) and Ag-LiveCodeBench-X (Boruch-Gruszecki et al., 2025) as evaluation benchmarks. MultiPL-E translates HumanEval (Chen et al., 2021) and MBPP (Austin et al., 2021) into over 19 languages, with MultiPL-HumanEval retaining 161 problems from the original set and MultiPL-MBPP retaining 396 problems. Ag-LiveCodeBench-X, derived from LiveCodeBench 5.0 (Jain et al., 2024), contains 499 problems and has a higher difficulty than MultiPL-E. We evaluate the low-resource PL Lua, along with C++ and Java, which represent PLs with complex syntax.

**Baselines.** We conduct comparative experiments on MultiPL-E and Ag-LiveCodeBench-X. First, we evaluate the performance improvement of *CodeChemist* compared to the original model (without test-time scaling). Then, under the same experimental setup, we compare *CodeChemist* with several representative test-time scaling strategies, including Majority Voting (Wang et al., 2023b), LLM Judge (Zheng et al., 2023a) (using 4o-mini as the judge model), Self-Debugging (Chen et al., 2023) and S* (Li et al., 2025a).

**Implementation Details.** We employ a multi-temperature hedged sampling strategy to generate $m = 10$ candidate solutions for each target PL problem. Specifically, the temperature is set to $t \in \{0.0, 0.7, 0.9, 1.1\}$, and 1, 3, 3, and 3 candidates are sampled in parallel, respectively. The initial

temperature for test case generation is set to 0.5, and the number of synthesized test cases is set to $n = 10$. For the uncertainty-aware selection, we set the clustering thresholds to $\delta_{ext} = 1$ and $\delta_{ast} = 2$, respectively. Detailed analysis of these hyperparameters is provided in Appendix B.2. Inference for the Qwen series and Llama3.2 is conducted locally on a single A100 GPU using the SGLang framework (Zheng et al., 2024), while 4o-mini and DeepSeek are accessed via their official APIs. All experimental code execution is performed on two Intel Xeon Platinum 8163 CPUs. All inference is performed using $top\text{-}p$=0.95, with the specific prompt details provided in Appendix D.

### 4.2. Main Results

Table 1 reports the comparison of *CodeChemist* on MultiPL-HumanEval against several methods: Vanilla (no test-time scaling), Majority Voting, LLM Judge, Self-Debugging and S*. The results demonstrate that *CodeChemist* consistently outperforms the baselines across most PLs and models, substantially enhancing the performance of low-resource PL Lua. Moreover, the larger the initial gap to high-resource PLs, the more pronounced the performance improvement in the low-resource PL. For example, on Qwen1.5B, the Python (63.9) vs. Lua (34.1) gap is close to 30.0 (see Appendix B.6), and *CodeChemist* achieves a 69.5% improvement on Lua compared with Vanilla. We provide additional baseline comparisons in Appendix B.4.

Across different target PLs, *CodeChemist* demonstrates the most pronounced improvements on the low-resource PL Lua, with relative gains ranging from 5.9% to 80.6%. For the C++ language, the improvements fall within 2.2%-51.7%, while for the Java language they lie within 4.9%-60.0%. Overall, *CodeChemist* consistently improves performance across all PLs, with particularly notable gains when the performance gap between PLs is larger. This trend highlights the extensibility of our method: it is effective not only on a typical low-resource PL like Lua, but also on PLs with complex syntax such as Java and C++.

Across different model families, *CodeChemist* delivers consistent performance gains, with the effect varying by model scale and PL disparity. For smaller models, where the performance gap between high- and low-resource languages is more pronounced, *CodeChemist* achieves the most significant improvements, thereby substantially enhancing their usability on target PLs. For example, on Qwen1.5B, the gains reach 69.5% for Lua, 51.7% for C++, and 60.0% for Java. For GPT-4o mini, although the performance across PLs is relatively close and the benefit from knowledge transfer is limited, *CodeChemist* still delivers gains of 7.1%, 5.5%, and 7.9%, effectively reducing the performance gap between high- and low-resource languages. For the strongest model, DeepSeek, performance across PLs is already rela-

*Table 1.* Pass@1 of Vanilla, majority voting, LLM judge, Self-Debugging, S*, and *CodeChemist* on MultiPL-HumanEval. The best performance is highlighted in bold, while the second best is underlined. Green arrows and values indicate improvements over the vanilla baseline, while red arrows and values denote a decrease in performance.

| Language | Method | Qwen 2.5 Coder Instruct | | | | Llama3 3B | GPT 4o -mini | DeepSeek V3 |
|---|---|---|---|---|---|---|---|---|
| | | 1.5B | 7B | 14B | 32B | | | |
| **Lua** | Vanilla | 34.1 | 69.7 | 74.2 | 78.0 | 29.9 | 74.8 | 82.1 |
| | Maj Voting | 45.3 ↑ 11.2 | 75.2 ↑ 5.5 | 77.0 ↑ 2.8 | 81.4 ↑ 3.4 | 46.6 ↑ 16.7 | 76.4 ↑ 1.6 | **88.2** ↑ 6.1 |
| | LLM Judge | 51.6 ↑ 17.5 | 73.9 ↑ 4.2 | 76.4 ↑ 2.2 | 77.0 ↓ 1.0 | 44.1 ↑ 14.2 | 75.2 ↑ 0.4 | 85.1 ↑ 3.0 |
| | Self-Debugging | 47.8 ↑ 13.7 | 74.5 ↑ 4.8 | 75.2 ↑ 1.0 | 81.4 ↑ 3.4 | 43.5 ↑ 13.6 | 77.0 ↑ 2.2 | 85.7 ↑ 3.6 |
| | S* | 50.9 ↑ 16.8 | 79.5 ↑ 9.8 | 75.8 ↑ 1.6 | 80.1 ↑ 2.1 | 50.9 ↑ 21.0 | 78.9 ↑ 4.1 | 82.0 ↓ 0.1 |
| | Ours | **57.8** ↑ 23.7 | **82.0** ↑ 12.3 | **80.8** ↑ 6.6 | **82.6** ↑ 4.6 | **54.0** ↑ 24.1 | **80.1** ↑ 5.3 | **88.2** ↑ 6.1 |
| **C++** | Vanilla | 34.4 | 73.0 | 77.5 | 83.9 | 37.5 | 80.1 | 93.0 |
| | Maj Voting | 49.1 ↑ 14.7 | 79.5 ↑ 6.5 | 82.6 ↑ 5.1 | 85.7 ↑ 1.8 | 52.8 ↑ 15.3 | 83.2 ↑ 3.1 | 92.6 ↓ 0.4 |
| | LLM Judge | 45.3 ↑ 10.9 | 80.1 ↑ 7.1 | 82.0 ↑ 4.5 | 85.7 ↑ 1.8 | 51.6 ↑ 14.1 | 80.8 ↑ 0.7 | 92.6 ↓ 0.4 |
| | Self-Debugging | 49.7 ↑ 15.3 | 68.9 ↓ 4.1 | 79.5 ↑ 2.0 | 85.1 ↑ 1.2 | 47.8 ↑ 10.3 | 78.9 ↓ 1.2 | 93.2 ↑ 0.2 |
| | S* | 46.0 ↑ 11.6 | 76.4 ↑ 3.4 | 82.0 ↑ 4.5 | 85.7 ↑ 1.8 | 53.4 ↑ 15.9 | 80.1 ↑ 0.0 | 93.2 ↑ 0.2 |
| | Ours | **55.3** ↑ 20.9 | **82.6** ↑ 9.6 | **85.7** ↑ 8.2 | **87.0** ↑ 3.1 | **54.0** ↑ 16.5 | **84.5** ↑ 4.4 | **95.7** ↑ 2.7 |
| **Java** | Vanilla | 43.5 | 77.7 | 81.5 | 81.5 | 37.9 | 79.6 | 89.3 |
| | Maj Voting | 62.7 ↑ 19.2 | 84.8 ↑ 7.1 | 83.5 ↑ 2.0 | 83.5 ↑ 2.0 | 53.8 ↑ 15.9 | 81.7 ↑ 2.1 | 91.8 ↑ 2.5 |
| | LLM Judge | 47.5 ↑ 4.0 | 79.1 ↑ 1.4 | 80.4 ↓ 1.1 | 84.2 ↑ 2.7 | 55.7 ↑ 17.8 | 79.1 ↓ 0.5 | 93.0 ↑ 3.7 |
| | Self-Debugging | 61.4 ↑ 17.9 | 78.5 ↑ 0.8 | 79.7 ↓ 1.8 | 84.2 ↑ 2.7 | 50.6 ↑ 12.7 | 80.1 ↑ 0.5 | 91.1 ↑ 1.8 |
| | S* | 67.1 ↑ 23.6 | 84.2 ↑ 6.5 | 82.3 ↑ 0.8 | 84.2 ↑ 2.7 | 59.5 ↑ 21.6 | 84.2 ↑ 4.6 | 90.5 ↑ 1.2 |
| | Ours | **70.3** ↑ 26.8 | **85.4** ↑ 7.7 | **86.7** ↑ 5.2 | **88.6** ↑ 7.1 | **59.5** ↑ 21.6 | **86.1** ↑ 6.3 | **93.7** ↑ 4.4 |

tively high, leaving limited room for further improvement. Nevertheless, *CodeChemist* still yields relative gains of 7.4%, 2.2%, and 4.9% on Lua, C++, and Java, respectively. This indicates that even in state-of-the-art models, cross-language knowledge transfer can play a complementary role, demonstrating the generality and robustness of the proposed method.

We showcase that our results are statistically significant via a t-test. More details are in Appendix B.1.

### 4.3. Results on Other Benchmarks

We further evaluate *CodeChemist* on MultiPL-MBPP and Ag-LiveCodeBench-X, with the results shown in Table 2. We compare *CodeChemist* against the aforementioned baselines. The experimental results further demonstrate that *CodeChemist* consistently outperforms these baseline methods across various benchmarks, models, and PLs.

On the relatively easier MultiPL-MBPP benchmark, *CodeChemist* achieves consistent and substantial performance gains. For example, on Qwen1.5B, Lua/Java/C++ improve by 56.9%/29.3%/43.7%, respectively; even for the powerful DeepSeek-V3, improvements in Lua/Java/C++ reached 13.7%/7.7%/13.9%, respectively.

Compared to MultiPL-MBPP, Ag-LiveCodeBench-X

presents a more challenging and realistic evaluation scenario. As shown in Figure 1, the baseline performance for Lua on this benchmark is suboptimal (2.8–36.5). However, *CodeChemist* achieves substantial relative improvements ranging from 18.0% to 200.0%, effectively enhancing the performance of low-resource PLs. Specifically, for Qwen-1.5B, which exhibits limited proficiency in Lua, *CodeChemist* more than doubles the performance. Even when applied to the state-of-the-art DeepSeek-V3, *CodeChemist* achieves a substantial relative gain of 38.4% in Lua. For C++ and Java, *CodeChemist* also provides consistent gains, with improvements of 7.3%-55.8% and 5.9%-107.1%, respectively, indicating its effectiveness even in tasks with higher algorithmic complexity and difficulty. We further extend the evaluation to additional low-resource PLs, including Julia and R, in Appendix B.5.1.

### 4.4. Ablation Studies

We perform ablation studies on *CodeChemist* on the MultiPL-HumanEval benchmark to analyze its key components, focusing on the contributions of the multi-temperature hedged sampling, test case generation strategies, and the adaptive selection mechanism.

**Sampling Strategy.** We use Pass@1 to measure the diversity of candidate pools and compare two schemes: (i)

*Table 2.* Pass@1 of Vanilla, majority voting, LLM judge, Self-Debugging, S*, and *CodeChemist* on MultiPL-MBPP and Ag-LiveCodeBench-X. The best performance is highlighted in bold, while the second best is underlined. Green arrows and values indicate improvements over the vanilla baseline, while red arrows and values denote a decrease in performance.

| Language | Method | MultiPL-MBPP | | | Ag-LiveCodeBench-X | | |
|---|---|---|---|---|---|---|---|
| | | Qwen (1.5B) | Qwen (32B) | DeepSeek-V3 | Qwen (1.5B) | Qwen (32B) | DeepSeek-V3 |
| Lua | Vanilla | 36.9 | 60.7 | 57.8 | 2.8 | 21.5 | 36.5 |
| | Maj Voting | 55.9↑19.0 | 63.2↑2.5 | 64.2↑6.4 | 6.2↑2.1 | 30.7↑9.2 | 47.3↑10.8 |
| | LLM Judge | 49.1↑12.2 | 61.0↑0.3 | 55.7↓2.1 | 4.2↑1.4 | 23.9↑2.4 | 38.9↑2.4 |
| | Self-Debugging | 53.1↑16.2 | 61.5↑0.8 | 64.0↑6.2 | 4.0↑1.2 | 28.1↑6.6 | 45.7↑9.2 |
| | S* | 50.4↑13.5 | 63.7↑3.0 | 62.7↑4.9 | 5.4↑2.6 | 31.2↑9.7 | 48.1↑11.6 |
| | **Ours** | **57.9**↑21.0 | **66.5**↑5.8 | **65.7**↑7.9 | **6.8**↑4.0 | **33.1**↑11.6 | **50.5**↑14.0 |
| C++ | Vanilla | 39.6 | 65.8 | 62.6 | 8.4 | 36.8 | 65.0 |
| | Maj Voting | 53.2↑13.6 | 65.7↓0.1 | 68.8↑6.2 | **11.2**↑2.8 | 38.9↑2.1 | 70.3↑5.3 |
| | LLM Judge | 55.4↑15.8 | 64.2↓1.6 | 68.0↑5.4 | 10.4↑2.0 | 38.1↑1.3 | 67.7↑2.7 |
| | Self-Debugging | 52.6↑13.0 | 66.5↑0.7 | 67.5↑4.9 | 8.2↓0.2 | 37.3↑0.5 | 71.5↑6.5 |
| | S* | 55.9↑16.3 | 68.0↑2.2 | 69.0↑6.4 | 10.8↑2.4 | 37.9↑1.1 | 70.5↑5.5 |
| | **Ours** | **56.9**↑17.3 | **68.8**↑3.0 | **71.3**↑8.7 | **11.2**↑2.8 | **40.9**↑4.1 | **72.3**↑7.3 |
| Java | Vanilla | 44.7 | 62.6 | 66.1 | 5.2 | 28.4 | 61.4 |
| | Maj Voting | 53.4↑8.7 | 66.8↑4.2 | 69.4↑3.3 | 7.2↑2.0 | **42.1**↑13.7 | 67.9↑6.5 |
| | LLM Judge | 51.3↑6.6 | 62.7↑0.1 | 63.5↓2.6 | 7.0↑1.8 | 32.5↑4.1 | 65.1↑3.7 |
| | Self-Debugging | 51.8↑7.1 | 64.2↑1.6 | 65.8↓0.3 | 5.0↓0.2 | 38.7↑10.3 | 68.1↑6.7 |
| | S* | 53.9↑9.2 | 66.3↑3.7 | 70.0↑3.9 | 7.0↑1.8 | 39.1↑10.7 | 68.7↑7.3 |
| | **Ours** | **57.8**↑13.1 | **67.9**↑5.3 | **71.2**↑5.1 | **7.4**↑2.2 | **42.1**↑13.7 | **71.3**↑9.9 |

*Table 3.* Comparing Pass@1 Scores on MultiPL-HumanEval using Qwen-32B: Single-Temperature Sampling (STS) vs. Multi-Temperature Hedged Sampling (MTHS).

| Sampling | Lua | C++ | Java | Avg. |
|---|---|---|---|---|
| STS (0.7) | 77.1 | **84.2** | 79.3 | 80.2 |
| STS (0.9) | 77.1 | 83.9 | 80.7 | 80.6 |
| STS (1.1) | 76.4 | 82.2 | **84.2** | 80.9 |
| **MTHS (Ours)** | **78.0** | 83.9 | 81.5 | **81.1** |

*Table 4.* Ablation study of test oracle generation on MultiPL-HumanEval, comparing different source languages (Python, C++ and Java) and high-resource PL sample sizes ($N = 1$ vs. $N = 10$). "Direct" denotes direct test oracle generation by the model without execution.

| Oracle | Qwen-1.5B | | | Qwen-32B | | |
|---|---|---|---|---|---|---|
| | Lua | C++ | Java | Lua | C++ | Java |
| Direct | 49.7 | 47.8 | 61.4 | 80.9 | 86.4 | 85.5 |
| C++(10) | 47.8 | 49.1 | 63.9 | 80.3 | 85.8 | 86.2 |
| Java(10) | 51.6 | 49.1 | 63.9 | 79.6 | 85.2 | 85.5 |
| Python(1) | 55.3 | 54.0 | 68.4 | 82.0 | 86.3 | 87.3 |
| Python(10) | **57.8** | **55.3** | **70.3** | **82.6** | **87.0** | **88.6** |

generating 10 samples with a fixed temperature of $\tau = 0.7$ (following the S* setting), $\tau = 0.9$, and $\tau = 1.1$; (ii) multi-temperature hedged sampling, which generates a single sample at $\tau = 0$ and 3 samples each at $\tau \in \{0.7, 0.9, 1.1\}$. The results on Qwen-32B are shown in Table 3. The experiments indicate that hedged sampling outperforms single-temperature sampling in most cases, highlighting the importance of balancing stability and diversity through the multi-temperature setting.

**Test Oracle Generation.** In the test case generation phase, we generate 10 samples from a high-resource PL and employ majority voting to construct test oracles. We hypothesize that both the choice of the source language and the number of generated samples significantly affect the accuracy of these oracles, which in turn impacts generation performance in target PLs. To validate this, we conduct a comparative analysis under three different settings: Direct Generation

(without execution); using other PLs as the source; and employing a single sample ($N = 1$) for test oracle generation. The results on Qwen-1.5B and Qwen-32B are presented in Table 4.

Impact of Source Language: Python proves to be the most effective source language, validating our hypothesis that the efficacy of functional transfer depends on the performance disparity between source and target PLs. For instance, on Qwen-1.5B, Python's superior vanilla performance (63.9) yields the highest transfer score on Lua (57.8), significantly outperforming Java (51.6) and C++ (47.8). This trend persists in Qwen-32B, confirming Python's role as a superior

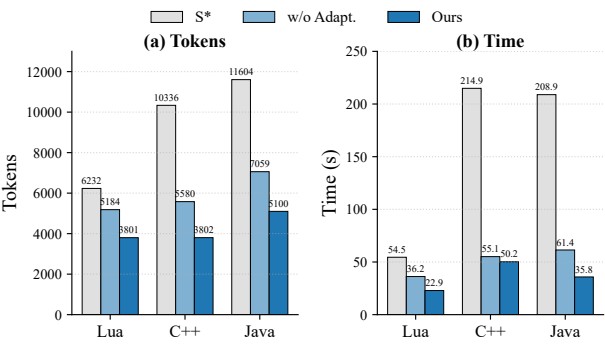

*Table 5.* Ablation study of the adaptive selection mechanism on MultiPL-HumanEval using Qwen-1.5B (Pass@1). "w/o Adapt." denotes the CodeChemist variant without uncertainty-aware adaptive selection.

| Method | Lua | C++ | Java |
|---|---|---|---|
| w/o Adapt. | **57.8** | 52.2 | 69.6 |
| Ours | **57.8** | 55.3 | 70.3 |

*Figure 3.* Comparison of computational efficiency on the MultiPL-HumanEval benchmark using Qwen-32B. "w/o Adapt." refers to the CodeChemist variant without the uncertainty-aware adaptive selection mechanism.

"teacher." Beyond pursuing optimal transfer performance, specific scenarios are often constrained to particular source-target language pairs. To demonstrate the versatility of *CodeChemist* in such constrained contexts, we provide an instance of migrating from C++ to Rust in Appendix B.5.2.

Impact of Voting Strategy: Direct test oracle generation is prone to hallucination (Jain et al., 2024), especially in smaller models. To mitigate this, we employ a multi-sample voting strategy ($N = 10$). The benefit is particularly evident on Qwen-1.5B (Lua), where voting achieves 57.8, significantly outperforming both the direct generation baseline (49.7) and single-sample decoding (55.3). These results confirm that voting effectively filters out the stochastic noise inherent in single-sample generation, thereby enhancing the reliability of functional transfer.

**Adaptive Selection Mechanism.** Table 5 shows that our uncertainty-aware selection outperforms the non-adaptive baseline. By triggering cross-lingual knowledge transfer only when necessary, it effectively mitigates the potential minor noise introduced by high-resource reference code (see Appendix C), leading to improved overall performance.

### 4.5. Computational Efficiency Analysis

We analyze computational efficiency in terms of token usage and time cost per problem. As shown in Figure 3, we compare three methods on Qwen-32B: $S^*$, w/o Adapt. (a non-adaptive variant that always triggers cross-lingual knowledge transfer), and Ours. Full comparisons with other baselines are deferred to Appendix B.3.

In terms of resource consumption, it consumes approximately **2× fewer tokens** and is **4× faster** than $S^*$, the second-best performing method. Compared to the variant without adaptive selection (w/o Adapt.), it also reduces both token usage and time cost by nearly **30%**. This efficiency gap primarily stems from algorithmic differences: whereas

$S^*$ relies on computationally expensive iterative LLM calls for adaptive input generation and pairwise comparisons, *CodeChemist* adopts an uncertainty-aware adaptive design. Specifically, smaller models with lower inference costs often exhibit high uncertainty. Conversely, larger models with higher inference costs demonstrate low uncertainty (see Appendix B.3). This reduces unnecessary costs for large models, while smaller models leverage additional compute to improve performance. Consequently, *CodeChemist* achieves a superior trade-off between accuracy and computational efficiency.

Furthermore, we explored methods to further minimize inference costs. For instance, employing the faster Qwen-3B to generate high-resource candidates for the Qwen-32B model improves Lua performance from 79.5 (Vanilla) to 81.4. Although this entails a slight compromise in accuracy, it significantly reduces inference overhead. Overall, these results confirm that *CodeChemist* effectively balances performance gains with computational efficiency, providing a practical solution for enhancing the performance of low-resource PLs.

### 4.6. Discussion

We further analyze *CodeChemist* from three complementary perspectives: (i) **Compatibility with Reasoning-Enhanced Models.** We evaluate *CodeChemist* on DeepSeek-V3 in Thinking Mode and observe a +5.0% relative improvement on Lua (see Appendix B.5.3), indicating strong compatibility with reasoning-enhanced models. (ii) **Combination with Other Test-Time Scaling Methods.** Since *CodeChemist* focuses on cross-language knowledge transfer via test case generation while methods such as S* primarily optimize selection within a single language, we combine *CodeChemist* with S* on MultiPL-HumanEval (see Appendix B.5.5) and achieve 83.9 Pass@1 on Qwen 7B (Lua), surpassing *CodeChemist* (82.0) and S* (79.5). (iii) **Impact on Code Style.** We further assess the impact of our method on coding style in Appendix B.7, where the generated code exhibits no significant stylistic degradation.

# 5. Conclusion

We propose *CodeChemist*, an uncertainty-aware test-time scaling framework that transfers functional knowledge from high-resource PLs to low-resource PLs through synthesized test cases. *CodeChemist* first applies multi-temperature hedged sampling to generate a pool of low-resource candidates and synthesize test inputs, and then estimates uncertainty by clustering candidates based on execution behavior and AST similarity. When uncertainty is low, *CodeChemist* selects the final output via in-language majority voting; otherwise, it constructs cross-lingual I/O test oracles by executing high-resource reference programs and selects the candidate with the highest pass rate. Extensive experiments on MultiPL-E and Ag-LiveCodeBench-X demonstrate that *CodeChemist* consistently improves code generation across both low-resource and complex-syntax PLs, outperforming existing test-time scaling methods, with particularly strong gains when the capability gap between languages is large.

# 6. Limitations

Despite its effectiveness, *CodeChemist* has several limitations that suggest directions for future work. First, *CodeChemist* is most applicable to scenarios where functional logic can be transferred across PLs, such as cross-platform development or legacy code migration. For tasks that heavily rely on language-specific mechanisms, especially those involving non-functional properties such as parallel efficiency and execution overhead, input-output behavior alone may be insufficient to fully characterize program semantics. As a result, the method remains limited in such scenarios. Second, *CodeChemist* is primarily designed to transfer functional behavior from high-resource PLs to low-resource PLs. For tasks that remain difficult to solve even in high-resource PLs, the effectiveness of this approach is correspondingly limited. Finally, our evaluation focuses mainly on function-level code generation tasks. Although our RepoClassBench (Deshpande et al., 2024) case study (see Appendix B.5.4) suggests potential beyond function-level tasks, broader validation on repository-level code generation remains future work.

# Acknowledgements

This research was partially supported by the Key Research and Development Project in Shaanxi Province No. 2024PT-ZCK-89, Ant Group, the National Natural Science Foundation of China Nos. 62406242, 62476215, 62302380, 62037001 and 62137002, and the Project of China Knowledge Centre for Engineering Science and Technology.

# Impact Statement

This paper aims to advance CodeLLMs for more reliable code generation across programming languages, especially for low-resource and complex-syntax Programming Languages. Potential societal impacts include improved developer productivity, broader accessibility to programming tools, and more consistent software quality through better cross-language support.

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

## A. Algorithm and Implementation Details

### A.1. Algorithm Implementation

The algorithm 1 presents the overall workflow of *CodeChemist*. First, *CodeChemist* applies multi-temperature hedged sampling to generate a pool of candidate programs in the target PL, and synthesizes a set of test inputs. Second, it estimates uncertainty by clustering candidates based on their execution behaviors and AST similarity. Finally, it performs adaptive selection: when uncertainty is low, it selects the output via in-language majority voting; otherwise, it constructs cross-lingual I/O test oracles by executing high-resource reference programs on the same inputs and selects the candidate with the highest pass rate.

---

**Algorithm 1** The Implementation of *CodeChemist*

---

**Input** : Problem $P$
**Output** : Best sample $x^*$

1   $X \leftarrow \text{MultiTempSampling}(P, \{\tau_1, \tau_2, \ldots, \tau_k\})$ ;     *// Hedged sampling*
2   $I \leftarrow \text{GenTests}(P)$ ;     *// Generate test inputs*
3   $C^{Ext} \leftarrow \text{ClusterByExecution}(X, I)$ ;     *// Cluster based on outputs*
4   $C^{AST} \leftarrow \text{ClusterByAST}(X)$ ;     *// Cluster based on syntax*
5   $k_{ext} \leftarrow |C^{Ext}|, \quad k_{ast} \leftarrow |C^{AST}|$
6   $S \leftarrow [0] \times |X|$ ;     *// Initialize score array*
7   **if** $k_{ext} \leq \delta_{ext} \wedge k_{ast} \leq \delta_{ast}$ **then**
     *// Low Uncertainty (Self-Voting)*
8      $x^* \leftarrow \text{Self-Voting}(X)$
9   **else**
     *// High Uncertainty (Knowledge Transfer)*
10      $H \leftarrow \text{GenHighCode}(P)$ ;     *// Generate high-resource reference*
11      $O \leftarrow []$ ;     *// Initialize Oracle outputs*
12      **for** $i_j \in I$ **do**
13        $R \leftarrow \{\text{Run}(h, i_j) \mid h \in H, \text{Valid}(h, i_j)\}$
14        **if** $R \neq \emptyset$ **then**
15          $O \leftarrow O \cup \{\text{MajorityVote}(R)\}$
16        **else**
17          $O \leftarrow O \cup \{\text{null}\}$
18      **for** $x_k \in X$ **do**
19        $p \leftarrow 0, v \leftarrow 0$ **for** $j \mid O[j] \neq null$ **do**
20          **if** $Run(x_k, I[j]) = O[j]$ **then**
21            $p \leftarrow p + 1$ ;     *// Pass count*
22          $v \leftarrow v + 1$ ;     *// Valid test count*
23        **if** $v > 0$ **then**
24          $S[k] \leftarrow p/v$
25        **else**
26          $S[k] \leftarrow 0$
27      $j^* \leftarrow \arg\max S$
28      $x^* \leftarrow x_{j^*}$ ;     *// Select the best sample*
29   **return** $x^*$ ;     *// Return best sample*

---

### A.2. Cross-Language Test Handling

Due to format differences across PLs, cross-language test handling requires input conversion and output normalization. Ag-LiveCodeBench-X already uses a standard stdin/stdout format, allowing test cases to be shared directly across languages. For MultiPL-E, we rely on benchmark-provided tools to convert test inputs into target-language formats. Outputs are handled by a unified normalization and comparison pipeline, which includes primitive standardization, complex-structure unfolding, and tolerance-based verification, as detailed below.

**Primitive Standardization:** To resolve cross-lingual representation ambiguities, we enforce a strict, type-aware format. For Numbers, we prefix values with `<N>` to ensure type safety and distinguish them from numeric strings (e.g., `1.23` → `<N>1.23`). For Booleans, we apply a type tag and normalize to lowercase to prevent casing discrepancies (e.g., `True` → `true`). For Strings, internal newlines are escaped to preserve stream boundaries.

**Complex Structure Unfolding:** We completely unpack containers to produce a simplified stream of values. For Sequences, we list all atomic elements in occurrence order (e.g., `[1, 2, 3]` → `1, 2, 3`). For Maps, we lay out keys and values alternately, sorted by key to ensure determinism (e.g., `{b: 2, a: 1}` → `a, 1, b, 2`).

**Tolerance-Based Verification:** To address floating-point precision discrepancies inherent in different PLs (e.g., Python vs. C++), we apply numerical tolerance. Lines tagged with `<N>` are compared using a threshold (e.g., $\epsilon = 10^{-4}$), while all other lines enforce an exact string match.

# B. Supplemental Experiments

## B.1. Hypothesis Testing

We demonstrate the statistical significance of our results through t-tests. Specifically, we repeat the full evaluation five times with different random seeds and conduct paired t-tests between *CodeChemist* and the corresponding vanilla baseline for each model setting (Qwen1.5B, Qwen7B, Qwen14B, and DeepSeek-V3). We find that for all these settings, the p-values are $< 0.05$.

## B.2. Hyperparameter Analysis

**Analysis of the Number of Candidate Solutions.** Following SFS (Light et al., 2025), we set the sampling budget to $m = 10$ in our experiments. To examine this choice, Figure 4 reports the Pass@k scores for the Qwen 7B model as the number of sampled candidate solutions increases. The results show that performance gains for the evaluated PLs diminish beyond 9–10 candidates, suggesting that $m = 10$ provides a good balance between computational cost and performance.

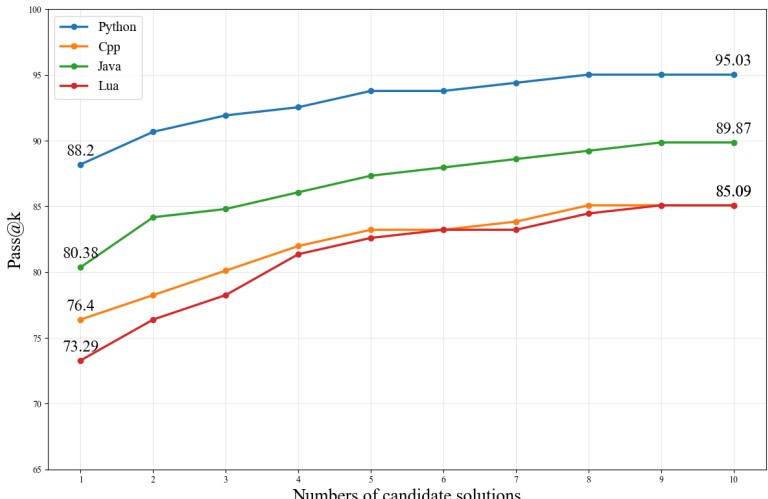

*Figure 4.* Analysis of the Number of Candidate Solutions for Qwen-7B on the MultiPL-HumanEval Benchmark.

**Analysis of the Test Case Quantity**. Similarly, for the number of synthesized test cases, we observed that a range of 8 to 10 is sufficient to effectively filter candidate solutions, while using fewer than 5 leads to insufficient discrimination. Therefore, we fixed $n = 10$ to strike a reasonable trade-off between selection accuracy and efficiency.

**Analysis of Clustering Thresholds.** We set the clustering thresholds to $\delta_{ext} = 1$ and $\delta_{ast} = 2$ to strictly define the high-confidence regime for our uncertainty-aware selection. For execution consistency, we enforce a strict constraint ($\delta_{ext} = 1$), requiring all sampled candidates to converge to a single execution outcome ($k_{ext} \leq 1$). This ensures that the model exhibits zero functional ambiguity regarding the program's behavior. In contrast, we set $\delta_{ast} = 2$ to tolerate minor structural variations in implementation that preserve the same logic. This accommodates legitimate diversity ($k_{ast} \leq 2$) while effectively filtering out chaotic outputs.

## B.3. Computational Efficiency Analysis

Table 6 reports how often the sampled candidate pool forms a clear majority-voting consensus (i.e., a dominant cluster) on MultiPL-HumanEval. We observe a strong scaling trend: smaller models exhibit low consensus rates, indicating high uncertainty and diverse generations, whereas larger models achieve substantially higher consensus. This motivates *CodeChemist*'s uncertainty-aware routing: when consensus is high, the method can rely on cheap in-language voting; when consensus is low, it invokes cross-lingual oracle construction to improve accuracy.

We analyze the computational cost of different test-time scaling methods in terms of token usage and wall-clock time per problem. As shown in Table 7, *CodeChemist* offers a favorable balance between efficiency and accuracy: although its cost is slightly higher than majority voting, it remains substantially more efficient than $S^*$. This advantage comes from our uncertainty-aware design. Specifically, *CodeChemist* triggers the expensive cross-lingual oracle construction only when the candidate pool exhibits high uncertainty; otherwise, it falls back to cheap in-language majority voting. Consequently, the method allocates computation where it is most needed, yielding strong empirical gains without incurring the consistently high overhead of fully iterative approaches.

*Table 6.* Percentage of successful consensus formation via majority voting on Lua, Java, and C++ in MultiPL-HumanEval.

| Benchmark | Qwen 2.5 Coder Instruct | | | | Llama3 3B | GPT 4o -mini | DeepSeek V3 |
|---|---|---|---|---|---|---|---|
| | 1.5B | 7B | 14B | 32B | | | |
| Lua | 7.46% | 42.24% | 52.17% | 60.87% | 5.59% | 40.37% | 71.43% |
| Java | 25.95% | 65.83% | 82.28% | 84.81% | 10.13% | 63.29% | 79.75% |
| C++ | 14.29% | 60.25% | 77.64% | 78.26% | 8.70% | 54.66% | 81.37% |

*Table 7.* Comparison of token usage and time cost (seconds per problem) across methods in MultiPL-HumanEval using Qwen-32B. **Ours** achieves a balanced trade-off.

| Method | Token Usage | | | | Time Cost (s) | | | |
|---|---|---|---|---|---|---|---|---|
| | **Lua** | **C++** | **Java** | **Avg.** | **Lua** | **C++** | **Java** | **Avg.** |
| Vanilla | 327.9 | 459.2 | 620.7 | 469.3 | 8.6 | 7.8 | 13.5 | 10.0 |
| Maj Voting | 2911.9 | 3307.7 | 4754.7 | 3658.1 | 14.3 | 48.8 | 33.8 | 32.3 |
| LLM Judge | 3915.3 | 5050.8 | 6995.6 | 5320.6 | 24.5 | 18.3 | 29.6 | 24.2 |
| Self-Debugging | 1221.0 | 1363.4 | 1836.9 | 1473.8 | 39.9 | 78.1 | 117.2 | 78.4 |
| S* | 6232.0 | 10335.5 | 11604.2 | 9390.6 | 54.5 | 214.9 | 208.9 | 159.5 |
| **Ours** | 3801.1 | 3801.7 | 5099.9 | 4234.2 | 22.9 | 50.2 | 35.8 | 36.3 |

## B.4. Comparison with Other Baselines

*Table 8.* Performance comparison across PLs on MultiPL-HumanEval.

| Model | Method | Lua | C++ | Java |
|---|---|---|---|---|
| | CodeT | 46.0 | 45.3 | 48.7 |
| Qwen-1.5B | MBR-Exec | 46.6 | 49.4 | 62.0 |
| | Ours | **57.8** | **55.3** | **70.3** |
| | CodeT | 81.4 | 85.7 | 85.4 |
| Qwen-32B | MBR-Exec | 81.4 | 85.1 | 84.8 |
| | Ours | **82.6** | **87.0** | **88.6** |

As shown in Table 8, our method outperforms CodeT (Chen et al., 2022a) and MBR-Exec (Shi et al., 2022) across all model scales and PLs. Notably, the performance of CodeT depends heavily on model-generated test assertions. For smaller models

such as Qwen-1.5B, these assertions tend to be noisy, causing CodeT to underperform MBR-Exec. In contrast, larger models such as Qwen-32B generate higher-quality assertions, allowing CodeT to slightly outperform MBR-Exec.

## B.5. Generalization Analyses

### B.5.1. EXTENSION TO MORE LOW-RESOURCE PL

We further extend *CodeChemist* to additional low-resource PLs, including Julia, which is widely used for scientific computing and physical simulation, and R, which is commonly used for statistical modeling. Results with Qwen-32B on Ag-LiveCodeBench-X show that *CodeChemist* consistently improves performance on both Julia and R, indicating its applicability to a broader range of low-resource code generation scenarios.

*Table 9.* Performance comparison on Ag-LiveCodeBench-X using Qwen-32B.

| Language | Vanilla | Ours | Rel. $\Delta$ (%) |
|----------|---------|-------|-------------------|
| R        | 12.32   | 21.84 | 77.27%            |
| Julia    | 31.08   | 37.27 | 19.92%            |

### B.5.2. BEYOND PYTHON AS THE SOURCE PL

*Table 10.* Pass@1 results when using C++ to improve Rust code generation on MultiPL-HumanEval.

| Model       | Vanilla | Ours | Rel. $\Delta$ (%) |
|-------------|---------|------|-------------------|
| Qwen-7B     | 74.2    | 80.8 | +8.9%             |
| DeepSeek-V3 | 89.7    | 92.3 | +2.9%             |

*CodeChemist* does not rely on Python as a specific source PL. To verify this, we conducted an additional experiment in which C++ serves as the teacher PL to enhance Rust code generation, as shown in Table 10. The results on MultiPL-HumanEval show that *CodeChemist* remains effective in this setting: the performance of Qwen-7B improves from 74.17 to 80.77 (+8.9%), and DeepSeek-V3 improves from 89.74 to 92.31 (+2.9%). These results provide initial evidence for the generality of the framework.

### B.5.3. EFFECTIVENESS UNDER DEEPSEEK THINKING MODE

*Table 11.* A Case of Thinking Mode in DeepSeek-V3 on MultiPL-HumanEval

| Language | Vanilla | Ours | Rel. $\Delta$ (%) |
|----------|---------|------|-------------------|
| C++      | 92.7    | 94.4 | +1.8%             |
| Java     | 92.5    | 93.7 | +1.2%             |
| Lua      | 84.6    | 88.8 | +5.0%             |

We evaluate the thinking mode of DeepSeek-V3 on the MultiPL-HumanEval benchmark, as shown in Table 11. The results show consistent improvements across PLs, with scores increasing from 92.73 to 94.41 for C++, from 92.53 to 93.67 for Java, and from 84.60 to 88.82 for Lua. These findings demonstrate that *CodeChemist* remains effective under the thinking mode setting.

### B.5.4. CASE STUDY ON REPOSITORY-LEVEL CLASS GENERATION.

To investigate whether *CodeChemist* can extend beyond function-level tasks, we conduct a case study on Task 56 of RepoClassBench. RepoClassBench evaluates class-level code generation under realistic repository contexts. When the contextual code exposes callable interfaces, *CodeChemist* can leverage cross-language interoperability tools, such as

pybind11 and pythonnet, to invoke the relevant context from the target-language repository. This allows *CodeChemist* to construct a Python reference program that reuses the actual repository context. *CodeChemist* then selects the candidate implementation whose behavior is most consistent with that of the Python reference program.

We evaluate this mechanism on Task 56 using candidate implementations generated by DeepSeek-V3. This task requires generating a C# class. *CodeChemist* successfully selects the correct implementation from the candidate pool, providing preliminary evidence that its functional knowledge transfer mechanism can extend to repository-level class synthesis. This case study serves as an initial feasibility validation, and we leave a systematic evaluation to future work.

### B.5.5. COMBINATION WITH OTHER TEST-TIME SCALING METHODS.

In this section, we explore how *CodeChemist* can be integrated with existing test-time scaling methods to enhance their performance. *CodeChemist* serves as a foundational framework designed to facilitate cross-language knowledge transfer through test case generation, which is distinct from traditional test-time scaling methods. While *CodeChemist*'s core focus is on generating high-quality, language-agnostic test cases to improve the performance of low-resource programming languages, other test-time scaling methods, such as S*, primarily concentrate on optimizing the candidate selection process within a single language.

We explore the combination of *CodeChemist* and the S* method. First, we generate a sample pool using high-temperature hedged sampling, and use Python to create language-agnostic test cases, followed by an initial filtering of the sample pool. Next, we compare the filtered samples pairwise, using LLM to generate inputs that can effectively distinguish between the two solutions. Then, we execute these adaptive inputs and provide feedback to the LLM based on the output, guiding it to make the optimal choice. In the Lua language experiment conducted on Qwen 2.5 Coder Instruct 7B, the performance improved from 69.7 to 83.9, further validating that *CodeChemist* can effectively combine with S* and significantly enhance the code generation capability for low-resource PLs.

### B.6. Performance of Python Vanilla Inference

Table 12 presents the Pass@1 results of Python native inference across three widely used benchmarks: MultiPL-HumanEval, MultiPL-MBPP, and Ag-LiveCodeBench-X. These results highlight the performance of different models, providing a basis for comparative analysis during the knowledge transfer process in *CodeChemist*.

*Table 12.* Pass@1 results of Python Vanilla inference performance across three benchmarks: MultiPL-HumanEval, MultiPL-MBPP, and Ag-LiveCodeBench-X.

| Benchmark | Qwen 2.5 Coder Instruct | | | | Llama3 3B | GPT 4o -mini | DeepSeek V3 |
|---|---|---|---|---|---|---|---|
| | 1.5B | 7B | 14B | 32B | | | |
| MultiPL-HumanEval | 63.9 | 87.6 | 89.2 | 91.9 | 57.8 | 87.7 | 93.0 |
| MultiPL-MBPP | 45.8 | 70.4 | 72.2 | 76.5 | 55.2 | 70.2 | 78.3 |
| Ag-LiveCodeBench-X | 7.7 | 20.9 | 31.7 | 36.4 | 17.0 | 50.0 | 70.1 |

### B.7. Impact on Code Style

Although Pass@1 is the primary metric used in this paper, code style is also important in practical applications. To assess whether *CodeChemist* affects the idiomaticity and stylistic quality of generated code, we use DeepSeek-V3 as an LLM-based judge and rate the generated solutions on a five-point Likert scale.

In the MultiPL-HumanEval experiments with Qwen-32B, *CodeChemist* and the Vanilla baseline obtain nearly identical scores, 4.276 and 4.288, respectively. This indicates that *CodeChemist* does not noticeably degrade code style.

## C. Failure analysis

Most incorrect test cases originate from incorrect Python reference programs. In these cases, other target languages are very likely to fail as well, with the conditional failure rate under the Vanilla method typically exceeding 90% (see Table 13). This suggests that *CodeChemist* is primarily bounded by the model's underlying problem-solving capability; the transferred test

cases mainly act as a medium for cross-language knowledge transfer.

*Table 13.* The conditional probability that target languages fail on Qwen-1.5B when the Python reference code is incorrect, under the Vanilla method.

| Language | MultiPL-HumanEval | Ag-LiveCodeBench-X |
|----------|-------------------|---------------------|
| Java | 0.78 | 0.96 |
| C++ | 0.91 | 0.94 |
| Lua | 0.91 | 0.97 |

## D. Prompts

In this appendix, we provide the detailed prompts used in our experiments. Our prompts are categorized by benchmark and by task type: (1) code generation and (2) test case generation. For reproducibility, we present the model's prompts.

### D.1. MultiPL-E

#### D.1.1. CODE GENERATION

> Example code generation prompt
>
> **Prompt:** Please continue to complete the function and return all completed code in a codeblock. Here is the given code to do completion:
> ```
> Question:{}
> ```

#### D.1.2. TEST CASE GENERATION

> Example Test Case Generation prompt
>
> **Prompt:** Please generate 10 diverse and meaningful test case inputs that thoroughly evaluate different aspects of the problem. Insert your test case inputs in the parentheses below and return only the code block:
> Question: {}
> ```
> # YOUR test case input HERE#
> ```

### D.2. Ag-LiveCodeBench-X

#### D.2.1. CODE GENERATION

> Example Code Generation prompt
>
> **Prompt:** You are a helpful assistant. You will be given a question (problem specification) and will generate a correct language program that matches the specification and passes all tests. You will NOT return anything except for the program.
> Question: {}
> Read the inputs from stdin solve the problem and write the answer to stdout (do not directly test on the sample inputs). Enclose your code within delimiters as follows.
> ```
> # YOUR CODE HERE#
> ```

D.2.2. TEST CASE GENERATION

---

Example Test Case Generation prompt

**Prompt:** You will be given a question (problem specification) and will generate 10 diverse and meaningful test case inputs that thoroughly evaluate different aspects of the question.
Problem: {}
Please read the input format carefully, directly return the generated test case, and do not generate code.
```
# YOUR test case input HERE#
```

---

