# OpenReview forum: "CodeChemist: Test-Time Scaling for Low-Resource Code Generation via Functional Knowledge Transfer"
_ICML.cc/2026/Conference — ICML 2026 regular_

### Official Review · Reviewer_U3yS · 2026-03-07

**Soundness:** 2
**Presentation:** 3
**Significance:** 2
**Originality:** 3
**Overall Recommendation:** 4
**Confidence:** 3

**Summary:**

This work introduces CodeChemist, a training-free test-time scaling framework that improves code generation for low-resource programming languages by transferring functional knowledge from high-resource ones. The system generates a pool of candidate solutions in the target language using multi-temperature hedged sampling, then estimates uncertainty by clustering candidates based on execution behavior and AST similarity. When uncertainty is low, it selects a final answer via majority voting; when uncertainty is high, it executes Python reference programs on synthesized test inputs to construct cross-lingual I/O oracles, selecting the candidate with the highest pass rate against them. Evaluated across multiple model families and benchmarks, CodeChemist consistently outperforms existing test-time scaling methods while using roughly 2x fewer tokens and running 4x faster than the next-best approach, with particularly strong gains on low-resource languages like Lua where the capability gap with Python is largest.

**Compliance With Llm Reviewing Policy:**

Affirmed.

**Final Justification:**

The authors have addressed my concerns during rebuttal and I have adjusted my score accordingly. My final recommendation is Weak Accept.

**Key Questions For Authors:**

See weaknesses.

**Limitations:**

I could not find a limitations section, please let me know if there is any.

**Strengths And Weaknesses:**

Strengths:
- Training free approach
- Efficiency
- Generalizability

Weaknesses:
- In Failure Analysis, you mentioned `Most incorrect test cases originate from incorrect Python reference programs`. What can be introduced to avoid this?
- Lack of other low resource languages. Why did you consider these three languages? What was the selection criteria?

---

> ### Author Rebuttal · Authors · 2026-03-31
>
> We thank the reviewer for the insightful review. Our responses are as follows:
> >W1: Mitigating incorrect Python reference programs.
>
>
>
> We thank the reviewer for this insightful comment. CodeChemist mitigates this issue through teacher-side consensus (Table 4) and adaptive selection (Table 5), as validated by the corresponding ablation studies.
> To further address this issue, we propose two extensions. First, we will adopt a multi-teacher transfer strategy that introduces additional high-resource languages when Python fails to reach consensus; preliminary experiments show that this increases Qwen-14B’s Pass@1 on Lua from 80.75 to 81.37. Second, we will integrate stronger single-language BoN methods to further improve test-case quality. The relevant content has been added to Appendix C (Line 827).
>
>
>
> >W2: Language selection criteria and other languages.
>
> Our language choices follow two complementary goals: using Lua to study low-resource settings, and using C++ and Java to evaluate generalization under greater syntactic complexity. CodeChemist is not tied to fixed language pair; we have also added a C++→Rust transfer case in Appendix B.5.
> To further broaden the language coverage, we extended the framework to R for statistical modeling and Julia for physical simulation. Results on Ag-LiveCodeBench-X with Qwen-32B show that the framework continues to deliver substantial gains on these languages. The relevant content has been added to Appendix B.5 (Line 784).
>
>
> | PL | Vanilla | Ours | Rel. Δ (%) |
> |---|---:|---:|---:|
> | R | 12.32 | 21.84 | 77.27% |
> | Julia | 31.08 | 37.27 | 19.92% |

---

> > ### Author Rebuttal · Reviewer_U3yS · 2026-04-01
> >
> > Thank you for addressing my concerns. I have updated my score and do not have additional questions.

---

> > > ### Author Response · Authors · 2026-04-01
> > >
> > > We are pleased to have addressed your concerns. We sincerely appreciate the time and effort you have taken to help improve the quality of our manuscript.

---

### Official Review · Reviewer_PXh3 · 2026-03-09

**Soundness:** 3
**Presentation:** 3
**Significance:** 3
**Originality:** 3
**Overall Recommendation:** 4
**Confidence:** 4

**Summary:**

To address the issue of large language models exhibiting weak coding capabilities in low-resource languages, this paper proposes a training-free test-time scaling method. First, it samples low-resource language code multiple times at different temperatures; in particular, the inclusion of a 1.1 temperature setting greatly enhances the diversity of the generated code. Subsequently, an uncertainty check is conducted by executing test inputs generated by the model. If the sampled low-resource codes exhibit high consistency in both semantics (output results) and syntax (AST), it indicates that the problem is simple for the model to solve in that language, and a final code is directly selected via majority voting. However, if the uncertainty of the sampled low-resource codes is high, the system enters the cross-lingual knowledge transfer module. In this module, the model samples high-resource code (such as Python) multiple times and executes the previously generated test inputs; when a specific input yields an absolute consensus among the outputs of these high-resource codes, that output is accepted as the truth. Because the functional knowledge of code (i.e., input-output mapping) is independent of programming languages, these ground-truth input-output pairs are ultimately used as samples to evaluate the low-resource code candidates, and the low-resource code with the highest score will win. Additionally, given the issue of inconsistent output formats across different languages, the authors unified a text-stream serialization protocol. Finally, the experimental section demonstrates the effectiveness and high efficiency of this method.

**Compliance With Llm Reviewing Policy:**

Affirmed.

**Ethical Review Concerns:**

The author has addressed some of my concerns; however, considering the overall quality of the paper, I have chosen to maintain my original rating.

**Final Justification:**

The author has addressed some of my concerns; however, considering the overall quality of the paper, I have chosen to maintain my original rating.

**Key Questions For Authors:**

See weaknesses.

**Limitations:**

Yes.

**Strengths And Weaknesses:**

### Strengths

1.The method proposed in this paper not only achieves SOTA (state-of-the-art) performance across various benchmarks but also has the highest computational efficiency. As shown in Figure 3, both its token consumption and time expenditure are the lowest. It possesses extremely high value for industrial applications.

2.The experimental design is thorough, with ablation studies conducted on various adjustable parameters, such as temperature selection and sampling strategy selection.


### Weaknesses

1.The evaluation metric is rather singular, relying solely on Pass@1. Particularly under high-temperature sampling conditions, the model may output relatively extreme code—for instance, code that merely passes the test cases but has a completely messy coding style.

2.The upper bound of this method is determined by the model's coding capability in high-resource languages. If the model cannot solve the problem in a high-resource language, it is even less likely to do so in a low-resource language. Future work could modify the paradigm for acquiring ground-truth input-output pairs to achieve the highest possible accuracy, which would be a highly challenging and constructive endeavor.

---

> ### Author Rebuttal · Authors · 2026-03-31
>
> We thank the reviewer for the positive feedback and constructive suggestions. Below, we address your insightful questions point by point.
>
> > W1: Evaluation metric.
>
>
> Thank you for raising this point. We use Pass@1 primarily to ensure fair comparisons with other methods (e.g., S* and Maj Voting). At the same time, we agree that code style is also important in practice. To assess code idiomaticity and stylistic quality, we use DeepSeek-V3 as an LLM-based judge and assign ratings on a five-point Likert scale. On Qwen-32B / HumanEval, Ours and Vanilla receive nearly identical scores (4.276 vs. 4.288), indicating that CodeChemist does not noticeably degrade code style; high-temperature sampling mainly expands the search space of algorithmic solutions. The relevant content has been added to Section 4.6 (Line 407).
>
>
>
>
> > W2:Upper bound of the method and future work.
>
> We thank the reviewer for this insightful comment. We agree that the upper bound of CodeChemist is inherently constrained by the model’s coding capability in the high-resource language. Our goal is not to solve tasks that remain unsolved in the high-resource language, but to transfer functional knowledge from the high-resource language to the low-resource language when such a capability gap exists. We also appreciate the reviewer’s suggestion that higher-quality ground-truth input-output pairs could further improve the framework’s upper bound. The relevant discussion has been added to Section 4.6 (Line 418).

---

> > ### Author Rebuttal · Reviewer_PXh3 · 2026-04-03
> >
> > I thank the reviewer for their response. The authors' reply has alleviated my concerns. However, I believe my previous rating remains justified; therefore, I will maintain my original score.

---

> > > ### Author Response · Authors · 2026-04-03
> > >
> > > We are glad that our response has alleviated your concerns. We greatly appreciate the thoughtful efforts you have dedicated to improving our manuscript.

---

### Official Review · Reviewer_HTs1 · 2026-03-13

**Soundness:** 4
**Presentation:** 3
**Significance:** 3
**Originality:** 3
**Overall Recommendation:** 4
**Confidence:** 4

**Summary:**

This paper studies test-time scaling(TTS) for low-resource code generation by cross-lingual code gen. The problem studied is an important problem of how to improve performance on low-resource or harder syntax target languages without additional training. The motivation of this work is straightforward: modern LLMs are quite strong in high-resource code languages like Python but also weak in low-resource code languages like Lua, which the author used as a test case. The paper proposed CodeChemist, a training-free TTS framework that transfers knowledge from a high-resource language in the target language using hedged multi-temperature sampling, then estimates the uncertainty by clustering candidates based on execution feedback and AST analysis. If uncertainty is low, the framework will directly select from the target language. And if the uncertainty is high, the stronger language code will be generated to build an input-output oracle, and the oracles will be used to rerank the target candidates. By this method, this paper treats the stronger language as a functional correctness teacher rather than a direct translation.

**Compliance With Llm Reviewing Policy:**

Affirmed.

**Final Justification:**

The rebuttal addressed my main concerns in a meaningful way. I especially appreciate the added clarification on the paper’s positioning and the additional evidence on Julia, R, and preliminary CUDA extension, as well as the clearer separation between adaptive routing and oracle-guided reranking. That said, my scope concern is only partially resolved, since the extension to more specialized domains is still limited and somewhat preliminary. Overall, the rebuttal strengthens the paper, and I will maintain my positive score of 4.

**Key Questions For Authors:**

1. Have you tried the methodology on some more important low-resource code gen domain? Like CUDA, physics modeling? A strong answer here will make me have much larger confidence in this work.

2. Can you more explicitly quantify how much of the gain comes from adaptive routing versus oracle-guided reranking itself?

**Limitations:**

yes

**Strengths And Weaknesses:**

Strength: The studied problem is important. The paper's TTS framework doesn't need additional low-resource data and training is free is very well motivated and useful. The framework is clean and straightforward, and the decomposition into hedged sampling, uncertainty estimation, and oracle-guided selection is easy to follow and also well motivated too. The empirical story is strong that gains from multiple languages models and benchmarks are reported and strong. The efficiency analysis is also valuable. The ablation coverage is also broad and good.

Weakness: My biggest concern is the scope of this paper. The author is trying to use strong language to set up input-output oracles for low-resource oracles. Actually, it is less clear how broadly this framework extends to some of the most important low-resource and specialized code-generation domains. In many such cases, there may be no natural high-resource source language that can describe the same functionality,  and no reliable oracle that can be easily constructed at test time. That's why low-resource code languages exist; they are solving different and really hard problems. Examples include hardware-specific code generation (e.g., CUDA/PTX-style kernels) and scientific computing or simulation code with complex dependencies and execution semantics.

The second concern is about the position of this work; this paper presents itself as improving target-language code generation, but the method actually relies on cross-lingual test-time assistance from a stronger language. That is a legitimate setting, but I think the paper should be very explicit that the improvement does not come purely from better target-language decoding, but from using stronger-language functionality as a teacher at inference time.

---

> ### Author Rebuttal · Authors · 2026-03-31
>
> We thank the reviewer for the positive comments and helpful suggestions. We address the comments below one by one.
>
> >W1: Scope and Applicability
>
>
>
> We thank the reviewer for this important comment. We agree that CodeChemist does not aim to cover all low-resource or specialized code-generation scenarios. Our goal is to address a broad and practical class of settings where the core logic is language-agnostic, while the final implementation language is constrained by engineering requirements, such as cross-platform development (e.g., iOS and Android), legacy code migration (e.g., C++→Rust), and multi-language library development (e.g., PyTorch and js-PyTorch). To further assess this scope, we extended the framework to more complex settings, including Julia for physical simulation, R for statistical modeling, and CUDA (see Q1). The relevant content has been added to Appendix B.5 (Line 784).
>
>
>
> >W2: The position of this work.
>
> We thank the reviewer for pointing this out. We agree that the positioning of our method should be stated more explicitly. In the revised manuscript, we clarify that the gain does not come purely from improved target-language decoding, but from cross-language functional transfer at inference time. The relevant discussion has been added to Section 1 (Line 89) and Section 3 (Line 157).
>
>
>
> >Q1: Extension to more low-resource domains.
>
> We thank the reviewer for highlighting the importance of evaluating the framework on more important low-resource code-generation domains. In addition to the C++→Rust results already reported in Table 8, we further extended CodeChemist to several important low-resource code-generation domains, including Julia for physical simulation and scientific computing, R for statistical modeling, and CUDA. On Ag-LiveCodeBench-X, results with Qwen-32B show that codeChemist continues to deliver consistent gains on Julia and R. On KernelBench, we also conducted a preliminary CUDA study: using an LLM-generated PyTorch implementation as reference logic, CodeChemist successfully identified a correct implementation among 20 CUDA candidates (demo: https://anonymous.4open.science/status/CodeChemist-cuda-25CC). The relevant content has been added to Appendix B.5 (Line 784).
>
>
>
>
> | PL | Vanilla | Ours | Rel. Δ (%) |
> |---|---:|---:|---:|
> | R | 12.32 | 21.84 | 77.27% |
> | Julia | 31.08 | 37.27 | 19.92% |
>
>
>
> >Q2: Contributions of adaptive routing and oracle-guided reranking.
>
>
> We apologize for not making this distinction sufficiently clear. Overall, the dominant accuracy gains come from oracle-guided reranking, while adaptive routing primarily improves efficiency by reducing unnecessary high-resource-language calls. As shown in Figure 3 and Table 6, adaptive routing skips high-resource assistance for a large fraction of problems; for example, with Qwen-32B on MultiPL-HumanEval, it avoids high-resource-language calls for 60.87% of Lua problems, 84.81% of Java problems, and 78.26% of C++ problems. In some settings, adaptive routing also yields a modest accuracy gain by avoiding noise propagation (e.g., C++ in Table 5: 52.2→55.3), but this gain does not appear in all settings. We have revised Section 4.4 (Line 406) to make this role separation more explicit.

---

> > ### Author Rebuttal · Reviewer_HTs1 · 2026-04-03
> >
> > The rebuttal addressed my main concerns in a meaningful way. I especially appreciate the added clarification on the paper’s positioning and the additional evidence on Julia, R, and preliminary CUDA extension, as well as the clearer separation between adaptive routing and oracle-guided reranking. That said, my scope concern is only partially resolved, since the extension to more specialized domains is still limited and somewhat preliminary. Overall, the rebuttal strengthens the paper, and I will maintain my positive score of 4.

---

> > > ### Author Response · Authors · 2026-04-03
> > >
> > > We are grateful that our response addressed your main concerns. We appreciate your feedback on the extension to specialized domains and will strengthen this discussion in the final version. Thank you again for your thoughtful and constructive feedback.

---

### Official Review · Reviewer_b67v · 2026-03-21

**Soundness:** 3
**Presentation:** 4
**Significance:** 2
**Originality:** 2
**Overall Recommendation:** 4
**Confidence:** 4

**Summary:**

This paper proposes an approach for natural language-to-function generation in low-resource programming languages where a codegen model is weaker (e.g. Lua) by generating reference functions in a language that is easier for the model (e.g. Python). The approach is an inference-time approach that samples many candidate functions in the low-resource language, and selecting a candidate that matches the reference functions' outputs as closely as possible on generated inputs. This pipeline is triggered when there is high disagreement among candidate outputs generated in the low-resource language (as measured by executing the functions on generated inputs, clustering the outputs, and checking the number of clusters), indicating model uncertainty. The paper evaluates on Lua, C++, and Java translations of HumanEval (from MultiPL-E) and LiveCodeBench (from Ag-LiveCodeBench-X) across a range of models (Qwen2.5, Llama 3, GPT-4o mini, and DeepSeek v3).

**Compliance With Llm Reviewing Policy:**

Affirmed.

**Final Justification:**

The rebuttal addressed my main concerns, and I raised my score to a 4 (from a 3).

**Key Questions For Authors:**

Questions:
- How are the inputs and outputs translated between languages in 3.4?
- How is AST clustering done? I understand, I think, the execution-based clustering. But AST would define a pairwise similarity between candidates, so it needs a clustering algorithm to produce clusters.
- Can you give more details on the majority voting? Does it use execution-based clustering (e.g. the same clustering done in 3.3, and choose a candidate from the large clusters? This would be similar to MBR-exec).
- How were the hyperparameters in the methods chosen (e.g. the cluster thresholds in 3.3), given that I don't think HumanEval and LiveCodeBench have development sets?

**Limitations:**

no; I couldn't find a limitation section. Please see the weaknesses above.

**Strengths And Weaknesses:**

Strengths:
- The paper presented generally thorough experiments, with ablations that showed the impact of various components.
- The approach shows consistent improvements over reasonable baselines.
- The paper was generally very clearly written.

Weaknesses
- The approach is only applicable for function-level generation and in settings where it is possible to translate inputs and outputs across languages. This is a good fit for the HumanEval and competition-style (LiveCodeBench) settings investigated here, which involve fairly simple data types, but it's not immediately clear how it would be applied to more complex and realistic settings.
- In many cases, the improvements of the full method over majority voting are small (e.g. on Ag-LiveCodeBench-X in Table 2, while Ours generally improves substantially over the Vanilla method, Maj Voting is often close to the performance of Ours, and nearly always obtains at least half of the performance improvement.) It's not clear that the extra complexity of generating code in the high resource language would be worth it.
- The method involves multiple hyperparameters and heuristics (the clustering thresholds in 3.3, the temperatures and number of candidates in the hedge sampling). See questions below.
- While the baselines were generally pretty reasonable, given that the method is generating inputs and candidate functions, it should also compare to CodeT (Chen et al. 2022) and MBR-Exec (Shi et al 2022) (both in the results, and the related work).
- The framing as "test-time-scaling" was a bit tenuous, IMO. The main novelty/focus of the paper is the generation of a reference in another language. But since it's sort of a test-time version of MultiPL-T / TransCoder-ST, I can see where this is coming from.

References
- Natural Language to Code Translation with Execution. Shi et al. 2022
- CodeT: Code Generation with Generated Tests. Chen et al. 2022

---

> ### Author Rebuttal · Authors · 2026-03-31
>
> We thank the reviewer for the insightful comments. Revisions have been incorporated into the manuscript.
>
> > W1: Code granularity and application scenarios.
>
> CodeChemist is not limited to function-level generation or simple data types. At the function level, CodeChemist transfers knowledge through I/O behavior and can be extended to other code-generation granularities that are verifiable via I/O. We also conducted a preliminary study on RepoClassBench for repository-level settings (demo: https://anonymous.4open.science/r/CodeChemist-Repo-4363). In terms of data types, our existing experiments already cover complex structures, such as C++ structs, Java queues, and Lua tables. More broadly, CodeChemist can be applied to scenarios that interact through cross-language serialization protocols (e.g., JSON and Protocol Buffers), such as microservice architectures.
>
> We focus on function-level code generation because it remains a mainstream and actively studied setting in current top-tier research [1,2]. Moreover, code generation itself is of broad practical importance, as it underlies many real-world software engineering tasks, such as feature implementation and algorithm development.
>
> [1] Execution Guided Line-by-Line Code Generation (NeurIPS25)
> [2] Sfs: Smarter code space search improves llm inference scaling (ICLR25)
>
> > W2: Cost-Effectiveness
>
> The additional cost yields proportionally higher effectiveness gains. In terms of accuracy, CodeChemist’s performance is driven by the cross-language capability gap. On Ag-LiveCodeBench-X, the smaller gap for Qwen (Table 10) leads to modest but consistent gains; for DeepSeek-V3, where the gap is larger, CodeChemist achieves substantial improvements.
>
> In terms of efficiency, CodeChemist acts as an on-demand intervention rather than adding constant complexity. Our uncertainty-aware router ensures that knowledge transfer is triggered only when necessary, thereby reducing unnecessary triggering in high-consensus scenarios (see Table 6). Consequently, our time and token costs remain close to Majority Voting (Table 7), whereas Majority Voting alone cannot exploit the cross-language capability gap to achieve similar gains.
>
> > W3: Hyperparameters.
>
> The details are provided in Q2 and Q4.
>
> > W4: Baselines
>
> We include CodeT and MBR-Exec in both the results section and the related work. Our method outperforms these baselines across all settings.
>
> | Model | Method | Lua | C++ | Java |
> |---|---|---:|---:|---:|
> | Qwen-1.5B | CodeT | 46.0 | 45.3 | 48.7 |
> |  | MBR-Exec | 46.6 | 49.4 | 62.0 |
> |  | Ours | __57.8__ | __55.3__ | __70.3__ |
> | Qwen-32B | CodeT | 81.4 | 85.7 | 85.4 |
> |  | MBR-Exec | 81.4 | 85.1 | 84.8 |
> |  | Ours | __82.6__ | __87.0__ | __88.6__ |
>
> > W5: Differences from other methods.
>
> MultiPL-T and TransCoder-ST primarily rely on translation, resulting in textual transfer, whereas our method focuses on functional transfer and places greater emphasis on semantic consistency. Textual transfer is often constrained by source-language syntax (e.g., translating Python into Rust may not naturally recover Rust ownership semantics) [1], while our functional-transfer paradigm supports more flexible transfer across diverse data types. In our experiments, this design better preserves idiomatic target-language expression, including Rust’s linear types and monadic operations, as well as C++’s zero-cost abstractions.
>
> [1] Program Skeletons for Automated Program Translation (PLDI 2025)
>
> > Q1: Details on cross-language interaction.
>
> We mainly rely on existing cross-language protocols and toolchains to handle cross-language interaction. Specifically, Ag-LiveCodeBench-X already uses a standard I/O format, so test cases are directly shared across languages. For MultiPL-E, we use tools provided by the benchmark to convert test cases into language-specific formats.
>
> > Q2: Details on AST clustering.
>
> We first compute the AST-distance matrix over candidate pairs using Tree Edit Distance, and then apply agglomerative hierarchical clustering to obtain the AST-based clusters and their counts used in Section 3.3.
>
> > Q3: Details on Majority Voting.
>
> Our majority voting is equivalent to MBR-Exec: candidates are clustered by full execution traces, and only candidates with identical outputs on all test cases are assigned to the same cluster.
>
> > Q4: Details on hyperparameters.
>
> These hyperparameters are empirically chosen and shared across benchmarks. Specifically, we fix the sampling budget at $m=10$ (following SFS), set $\delta_{ext}=1$ for strict execution consistency, and set $\delta_{ast}=2$ to allow minor structural differences. We thank the reviewer for pointing this out. We acknowledge that this shared configuration may be suboptimal for certain benchmarks. To further assess this issue, we constructed a development set from LiveCodeBench v4 and observed that increasing $m$ from 10 to 16 improves Ag-LiveCodeBench-X from 33.1 to 33.9.

---

> > ### Author Rebuttal · Reviewer_b67v · 2026-04-04
> >
> > Thanks to the authors for the response which addressed my main concerns.
> >
> > I'd appreciate more details about the RepoClassBench experiments in the revised version of the paper.
> >
> > Thanks too for the MBR-Exec and CodeT experiments! It's a bit surprising to me that CodeT does not outperform MBR-Exec and so I'd recommend checking these results.
> >
> > I'll update my score to a 4.

---

> > > ### Author Response · Authors · 2026-04-04
> > >
> > > We sincerely thank the reviewer for the continued and constructive feedback, which has been highly valuable in improving our manuscript. We will follow your suggestion to provide more detailed descriptions of the RepoClassBench experiments.
> > >
> > > Regarding the observation that CodeT does not outperform MBR-Exec, we also find it both surprising and insightful, and we appreciate you highlighting it. We conducted additional analysis and found a possible explanation: CodeT relies heavily on model-generated test assertions, while smaller models (e.g., Qwen-1.5B) tend to produce noisier assertions. In contrast, for larger models such as Qwen-32B, the improved quality of generated test assertions allows CodeT to slightly outperform MBR-Exec. We will incorporate these findings into the revised manuscript.

---

### Decision · Program_Chairs · 2026-04-30

**Decision:**

Accept (regular)

**Comment:**

CodeChemist proposes a training-free test-time scaling framework that transfers functional knowledge from high-resource to low-resource programming languages via cross-lingual I/O oracles, with strong efficiency and consistent gains across models and benchmarks. All four reviewers converged to weak accept (4/4/4/4) with concerns largely resolved; rebuttal added MBR-Exec/CodeT comparisons, Julia/R/CUDA extensions, code style evaluation, and clearer positioning of the cross-lingual transfer paradigm. Remaining scope concerns about specialized domains like CUDA are acknowledged but appropriately limited.